# Biocatalytic Oxidation of Alcohols

**Hendrik Puetz [1]**, **Eva Puchľová [2]**, **Kvetoslava Vranková [2]** and **Frank Hollmann [1,*]**

[1] Department of Biotechnology, Delft University of Technology, van der Maasweg 9, 2629 HZ Delft, The Netherlands; Hendrik.pue@live.de

[2] Axxence Slovakia s.r.o, Mickiewiczova 9, 81107 Bratislava, Slovakia; evka.puchlova@axxence.sk (E.P.); kvetka.vrankova@axxence.sk (K.V.)

**\*** Correspondence: f.hollmann@tudelft.nl

**Abstract:** Enzymatic methods for the oxidation of alcohols are critically reviewed. Dehydrogenases and oxidases are the most prominent biocatalysts, enabling the selective oxidation of primary alcohols into aldehydes or acids. In the case of secondary alcohols, region and/or enantioselective oxidation is possible. In this contribution, we outline the current state-of-the-art and discuss current limitations and promising solutions.

**Keywords:** alcohols; alcohol oxidation; alcohol dehydrogenases; alcohol oxidases; kinetic resolution; deracemization

## 1. Introduction

### 1.1. Why Use Biocatalysis for Alcohol Oxidations

The (catalytic) oxidation of alcohols is a mature yet still very active field of chemical research. Within the scope of Green Chemistry, well-established textbook methods are substituted with more efficient catalytic methods, shifting from problematic oxidants such as $AgNO_3$, $K_2CrO_4$ or $KMnO_4$ to environmentally more acceptable oxidants such as $O_2$ or $H_2O_2$ [1,2].

Biocatalysis could play a major role in this transition. Arguments frequently used in favor of biocatalysis are the mild reaction conditions and the renewable origin and biodegradability of enzymes. More importantly, however, enzymes are very selective catalysts enabling precision chemistry avoiding tedious protection group chemistry. At the same time, large parts of the chemical community tend to ignore enzymes as potential tools for synthesis planning, which is due to perceived and real limitations of enzyme catalysis.

In this contribution, we will briefly outline the current state-of-the-art in biocatalytic alcohol oxidations, highlighting synthetic opportunities but also critically discussing current limitations.

### 1.2. Biocatalysis for Alcohol Oxidation: Perceived and Real Limitations

Arguments frequently held against biocatalysis in general are its limited availability, narrow product scope, poor stability of the catalysts, and high price [3]. While this situation may have been true two decades ago, there has been tremendous progress alleviating or even solving many of the issues held against enzymes. Some of these will be discussed in the following sections.

#### 1.2.1. Availability of Oxidative Enzymes

Some 30 years ago, oxidative biocatalysis was largely restricted to natural diversity, i.e., enzymes available from natural resources. The famous alcohol dehydrogenase from horse liver (HLADH) [4,5] is just one prominent example of this. Then, HLADH was indeed obtained from horse liver resulting

in ethical issues and variations of availability and quality. With the rise of recombinant protein expression technology, cost-efficient and scalable enzyme production has become possible [6–8]. Various commercial enzyme suppliers offer oxidoreductases in quantities ranging from small to bulk.

In addition, the diversity of natural enzymes has been increasing (and continues doing so) considerably with new oxidative enzymes identified from metagenome libraries [9–12] and new habitats and organisms [13–16].

### 1.2.2. Substrate Scope

Natural enzymes have been optimized by natural evolution to serve the host organisms' purpose, which does not necessarily coincide with the needs of an organic chemist aiming at the selective oxidation of a given target molecule. Next to screening natural diversity for more suitable enzymes, protein engineering has become a very powerful tool to tailor the properties of a given enzyme such as cofactor specificity, thermo and solvent stability, (enantio)selectivity, and more [17–33].

For example, vanillyl alcohol oxidase has been engineered intensively by the groups of van Berkel and Fraaije to e.g., engineer the substrate specificity or the stereoselectivity of the hydroxylation reaction [34–38].

Another nice example exemplifying the power of protein engineering comes from the Alcalde lab [39]. Here, the aryl alcohol oxidase is from *Pleurotus eryngii*. Directed evolution resulted in enzyme mutants with higher stability and activity but also increased expression levels [21,40]. The wild-type enzyme shows only little activity toward secondary benzylic alcohols, which can be overcome by semi-rational design [20,41]; the resulting oxidase mutants were highly selective in the kinetic resolution of a range of secondary benzyl alcohols.

### 1.2.3. Stability

Compared to common chemical catalysts, the thermal stability of biocatalysts indeed is generally much lower. However, considering the high catalytic efficiency of enzymes at temperatures below 100 °C, the question arises as to why this should be an issue at all. High temperatures are generally applied to accelerate chemical transformations. However, a (bio)catalytic reaction proceeds sufficiently fast already; at more ambient temperatures, highly thermostable catalysts (operating at temperature ranges between 100 and 500 °C as commonly applied in chemical transformations) are not necessary.

Nevertheless, if activity and stability at elevated temperatures is desired, oxidoreductases from (hyper)thermophilic host organisms [13] such as *Pyrococcus* [42], *Thermus* [43,44], or *Sulfolobus* [45] are available.

### 1.2.4. Biocatalyst Costs

Finally, the seemingly high costs are spuriously held against enzymes. If purchased from a specialty chemical supplier, enzymes are indeed very expensive due to the usually small production scale. However, it should be kept in mind that enzyme production costs are subject to economy of scale and typical enzyme costs if produced at large scale are as low as 250 € $kg^{-1}$ enzyme [46]! A simple calculation reveals the catalyst performance needed to achieve a given cost contribution of the enzyme to the final product (Figure 1). For example, using an enzyme cost of 250 € $kg^{-1}$, only 100,000 turnovers are needed to attain an enzyme cost contribution of less than 1 € $kg^{-1}_{product}$. Obviously, this is an over-simplistic view, and other factors contribute to the cost structure of a given production process.

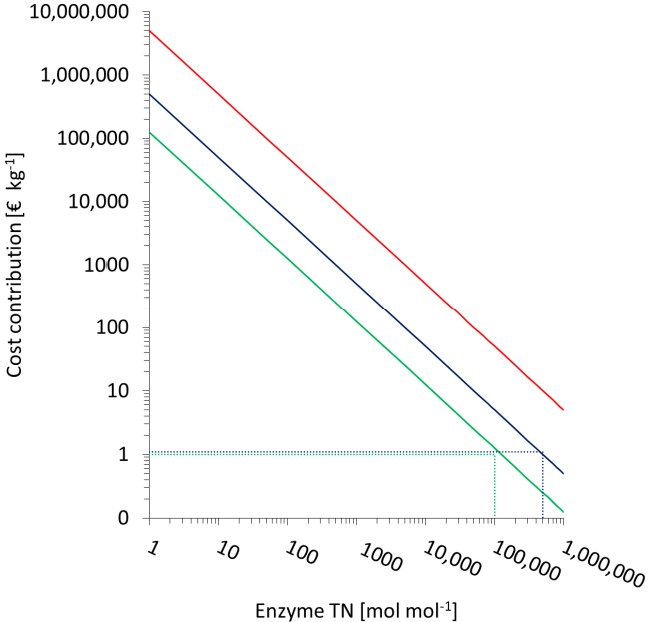

**Figure 1.** Cost contribution of a biocatalyst to the final product depending on the number of catalytic cycles performed (turnover number, TN = $mol_{product} \times mol^{-1}_{enzyme}$). Assumptions: molecular weight of the product: $200\,g \times mol^{-1}$; molecular weight of the enzyme: 200 kDa, **green**: enzyme costs = $250\,€\,kg^{-1}$, **blue**: enzyme costs = $1000\,€\,kg^{-1}$, **red**: enzyme costs = $10,000\,€\,kg^{-1}$.

Amongst the real limitations of biocatalysis is the still very common use of aqueous reaction mixtures. As the majority of reactants of interest are rather hydrophobic, aqueous reaction media support only concentrations in the lower millimolar range. Such low reagent concentrations are very unattractive from an economical point of view (Figure 2) as they also imply high operational costs (and follow-up cost for downstream processing handling of large volumes). Furthermore, large amounts of contaminated waste water will be generated, which have to be treated prior release into the environment, causing further costs and consuming energy and resources.

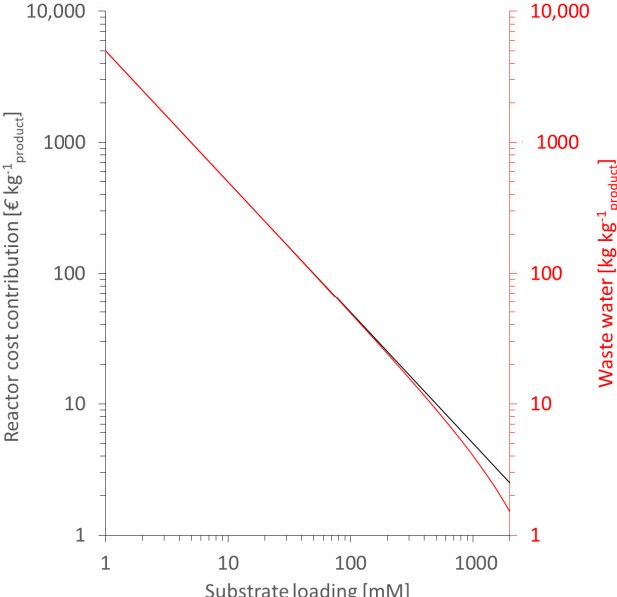

**Figure 2.** Estimation of the cost contribution of a reactor cost contribution (**black**) and waste water formed (**red**, based on a simple E-factor calculation). Assumptions: full conversion of the starting material into the product (Mw = $200\,g\,mol^{-1}$) and generic reactor costs of $1\,€\,L^{-1}$ reaction volume.

Fortunately, concepts such as non-aqueous or biphasic reaction mixtures are increasingly used, rendering biocatalytic oxidations more attractive from a preparative point of view.

*1.3. The Catalysts Used*

Various enzyme systems are available for the oxidation of alcohols. Next to the widely used alcohol dehydrogenases (ADHs) and alcohol oxidases (AlcOxs), so-called laccase-mediator systems (LMS) are worth mentioning also.

Alcohol dehydrogenases (also frequently denoted as ketoreductases, KREDs) utilize the oxidized nicotinamide cofactors (NAD(P)$^+$) as a hydride acceptor for the oxidation of alcohols (Scheme 1). The catalytic mechanism starts with the (reversible) binding of the oxidized nicotinamide cofactor (not shown) followed by the (likewise reversible) coordination of the alcohol starting material to the enzyme active site (Scheme 1, step (1)) and its deprotonation [47]. Coordination of the alcohol and NAD(P)$^+$ to a metal ion (often Zn$^{2+}$) ensures precise positioning of the alcohol–C–H bond to the pyridinium ring of NAD(P)$^+$ facilitating the hydride transfer (Scheme 1, step (2)). Finally, the carbonyl product leaves the enzyme active site (Scheme 1, step (3)), leaving the bound, reduced NAD(P)H behind (which can diffuse out of the enzyme active site or stay for a reductive round, which is the reversal of the just described oxidation reaction).

**Scheme 1.** Simplified mechanism of alcohol dehydrogenases (ADH)-catalyzed alcohol oxidation (**a**) all reaction steps are fully reversible. (**b**) Structure of the oxidized nicotinamide cofactors (NAD$^+$ and NADP$^+$).

The reversibility of the single reaction steps also explains the fact that ADHs can be used in both directions. In fact, the majority of ADH reports deal with the (enantioselective) reduction of prochiral ketones [27,48].

Cofactor Regeneration Strategies

The catalytic mechanism shown in Scheme 1 also implies that the oxidation of one equivalent of alcohol also results in the consumption of one equivalent of the oxidized nicotinamide (NAD(P)$^+$) yielding its reduced form (NAD(P)H). The still relatively high costs and frequently observed inhibitory effects (by both the oxidized and the reduced cofactor) on the biocatalysts prohibit their use in stoichiometric amounts. Therefore, over the years, a range of in situ NAD(P)$^+$ regeneration methods have been developed (Table 1). In essence, they allow using the costly nicotinamide cofactor in catalytic

amounts only as its active, oxidized form is continuously regenerated at the expense of a cosubstrate being reduced. In principle, two regeneration approaches can be distinguished: (1) the so-called substrate-coupled approach and the (2) enzyme-coupled approach.

The substrate-coupled approach exploits the reversibility of the ADH-catalyzed oxidation reaction by using the production ADH for NAD(P)$^+$ regeneration (driven by the ADH-catalyzed reduction of a cosubstrate, as shown in Figure 3a). Overall, this approach represents a biocatalytic variant of the chemical Oppenauer oxidation [49–52].

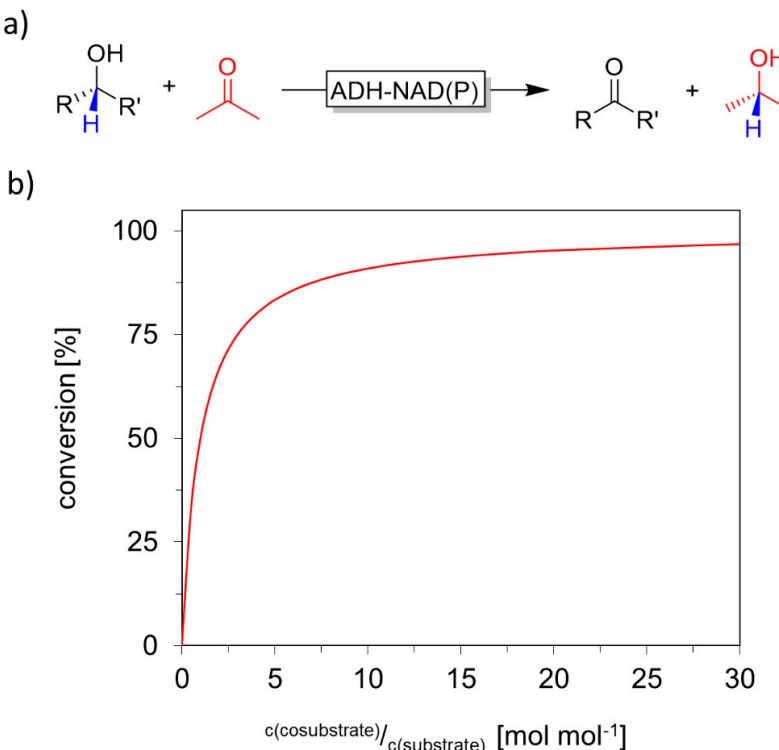

**Figure 3.** Substrate-coupled oxidation of alcohols. (**a**) Overall reaction using acetone as oxidant; (**b**) dependency of the equilibrium substrate conversion on the molar surplus of the cosubstrate (acetone) assuming an equilibrium constant K of 1.

Advantages of the substrate-coupled alcohol oxidation approach are that (1) the production enzyme also serves as a regeneration enzyme (no need for a second NAD(P)$^+$ regeneration catalyst), (2) the nicotinamide cofactor does not have to leave the enzyme active site for regeneration and thereby is less exposed to buffer-related degradation [53]. This also implies that ADH-catalyzed oxidation can principally be performed under non-aqueous conditions.

A disadvantage of the substrate-coupled approach is the low thermodynamic driving force of the overall reaction as the chemical composition of the products (alcohol and ketone) is essentially the same as that of the starting materials (alcohol and ketone). As a result of this, the equilibrium of the reaction is rather unfavorable, and additional measures are needed to shift the equilibrium. In some cases, removing one of the reaction products via extraction or distillation is possible. However, more common is to supply the (usually cheaper) cosubstrate in excess (Figure 3b). On the one hand, the surplus cosubstrate can be seen as a cosolvent, facilitating the solubilization of hydrophobic reagents. However, on the other hand, this surplus also represents an environmental burden and generates additional wastes [54].

A promising solution has been proposed by Kroutil and coworkers by using α-halo ketones as cosubstrates (Scheme 2) [55]. For example, chloroacetone enabled the authors to quasi-irreversibly oxidize a range of racemic alcohols using just 1.5 equivalents of the 'smart cosubstrate'. The authors

hypothesized that the coproduct was stabilized by intramolecular hydrogen bonding, thereby shifting the equilibrium. Unfortunately, chloroacetone is a strong lachrymator (e.g., used in pepper spray), rendering it unattractive for many applications (such as the synthesis of consumer products).

**Scheme 2.** One-way oxidation of alcohols using the ADH from Sphingobium yanoikuyae (SyADH) as a non-selective biocatalyst and chloroacetone as a 'smart cosubstrate'.

The enzyme-coupled regeneration approach relies on the cooperation of two catalysts, the $NAD(P)^+$-dependent ADH (production enzyme) and an $NAD(P)^+$-regenerating catalyst. At first sight, this approach seems more complicated than the above-discussed substrate-coupled approach. However, it allows us to make use of molecular oxygen as a terminal electron acceptor, thereby making use of the high thermodynamic driving force of oxygen reduction (Table 1). Hydrogen peroxide (which is generally dismutated into $O_2$ and $H_2O$ by the addition of a catalase) or water is formed as a by-product, which from a waste perspective is very attractive. To facilitate the aerobic oxidation of NAD(P)H, a range of enzymatic and non-enzymatic systems have been developed (Table 1*).*

For example, NADH oxidases directly re-oxidize NADH into $NAD^+$ while reducing molecular oxygen to $H_2O_2$ or $H_2O$ [56–64].

The so-called laccase mediator systems (LMSs) comprise combinations of laccases and chemical redox dyes to aerobically regenerate $NAD(P)^+$ from NAD(P)H. Compared to the aforementioned NADH oxidases, LMSs excel by their indifference with respect to the cofactor ($NADP^+$ or $NAD^+$) regenerated, since the NAD(P)H oxidation step is performed by an unselective chemical mediator (Scheme 3) [65–71]. Quite often, the reduced mediator itself reacts with molecular oxygen, making co-catalysis by laccase superfluous [72–77].

**Table 1.** Selection of aerobic $NAD(P)^+$ regeneration systems to drive ADH-catalyzed oxidation reactions.

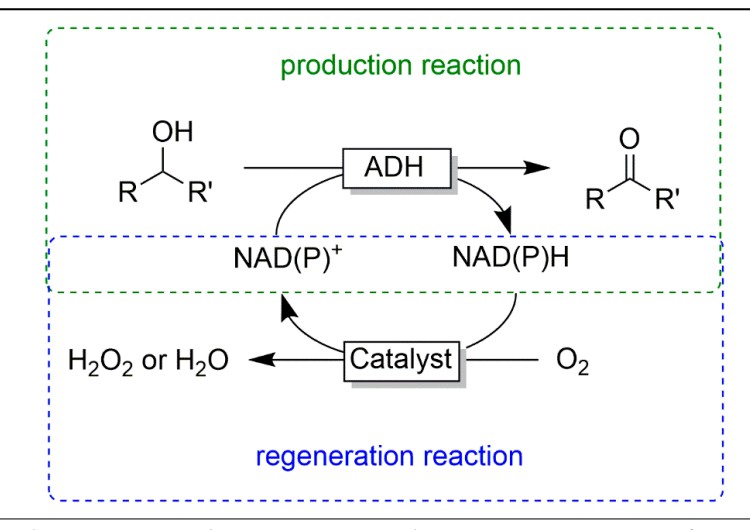

| Cosubstrate | Coproduct | Catalyst | Ref. |
|---|---|---|---|
| $O_2$ | $H_2O_2$ | (Modified) flavins | [72–77] |
| | $H_2O$ | LMS | [65–71] |
| | | NADH oxidase | [56,57,61,63,64,78–81] |
| | | Other oxidative enzymes | [82–86] |

LMS: laccase mediator system.

**Scheme 3.** Electron transport chain of a Laccase-mediator-system-promoted alcohol oxidation. The ADH catalyzes the NAD(P)$^+$-dependent oxidation reaction, yielding (NAD(P)H). The latter is spontaneously oxidized by a chemical mediator (a selection is shown on the bottom). The oxidized form of the mediator itself is re-formed by laccase-mediated aerobic oxidation.

Using molecular oxygen as a terminal electron acceptor is attractive from a thermodynamic driving force point of view as well as an environmental point of view, as only water is formed as a by-product. However, a major challenge of this approach originates from the very poor solubility of O$_2$ in aqueous media (under ambient conditions approximately 0.2–0.25 mM). As a consequence, the O$_2$ pool available is consumed, rapidly necessitating external provision with O$_2$ to drive the oxidation reaction [78,83,87,88]. Oxygen supply by bubbling air or O$_2$ through the reaction mixture represents a straightforward solution. The oxygen transfer rate is proportional to the gas–liquid interface area, which means that in principle, heavy sparking with small bubbles should be advantageous. However, many enzymes are inactivated at the liquid–gas interface [89–91]. A generally applicable solution is still elusive. It is also worth mentioning that strict regulations apply e.g., in the preparation of products used in natural foodstuffs and consumer care, excluding pressurized reactor setups.

Alcohol oxidases represent the second preparatively relevant class of enzymes useful for the oxidation of alcohols. In contrast to ADHs, AlcOXs do not rely on the nicotinamide cofactors but transfer the reducing equivalents liberated in the course of the alcohol oxidation reaction to molecular oxygen, yielding H$_2$O$_2$ as a stoichiometric by-product. Two main classes of AlcOxs are predominantly investigated: flavin-dependent AlcOxs and Cu$^{2+}$-dependent AlcOxs. As shown in Schemes 4 and 5, they differ significantly with respect to the oxidation mechanism.

**Scheme 4.** Simplified mechanism of flavin-dependent oxidase-catalyzed oxidation of alcohols. The alcohol starting material binds to the enzyme active site and is oxidized by deprotonation/hydride transfer to the oxidized flavin cofactor. The resulting reduced flavin cofactor re-oxidized by molecular oxygen is a cascade of single electron transfer steps.

**Scheme 5.** Simplified mechanism of Cu-dependent oxidases. $O_2$ binds to the ($Cu^+$) resting state of the enzyme, resulting in a double oxidation by simultaneous electron transfer from $Cu^+$ and H-atom abstraction from a phenolic active site amino acid. Next, the $Cu^+$-bound $H_2O_2$ is substituted by the alcohol starting material which undergoes H-atom abstraction (reforming the phenolic amino acid) and electron transfer to the $Cu^{2+}$ site, resulting in a $Cu^+$-coordinated carbonyl product that diffuses out of the enzyme active site to close the catalytic cycle.

In flavin-dependent oxidases, an oxidized flavin-cofactor abstracts the hydride from the alcohol C–H bond, yielding a reduced flavin. The oxidized cofactor is regenerated by re-oxidation with $O_2$ via a complex sequence of electron transfer steps, eventually yielding $H_2O_2$ as the by-product [92,93].

Cu-dependent oxidases follow a mechanism wherein a $Cu^{2+}$/phenoxy radical pair binds the alcohol starting material followed by a hydrogen atom abstraction step (to the phenoxy radical), yielding

the coordinated carbonyl product [94,95]. After dissociation of the product from the enzyme-active site, the reactive form is restored via $O_2$ reduction to $H_2O_2$.

Laccase-mediator systems essentially are organocatalytic alcohol oxidation reactions using oxammonium species as an oxidant. The stable *N*-oxide radical TEMPO (or its analogues) is oxidized by the blue-copper enzyme laccase (at the expense of $O_2$ being reduced to $H_2O$) to the reactive oxammonium special, forming a covalent adduct with the alcohol starting material. After the oxidation step, a hydroxylamine is formed that synproportionates with another oxammonium molecule, forming the TEMPO catalyst (Scheme 6) [96–99].

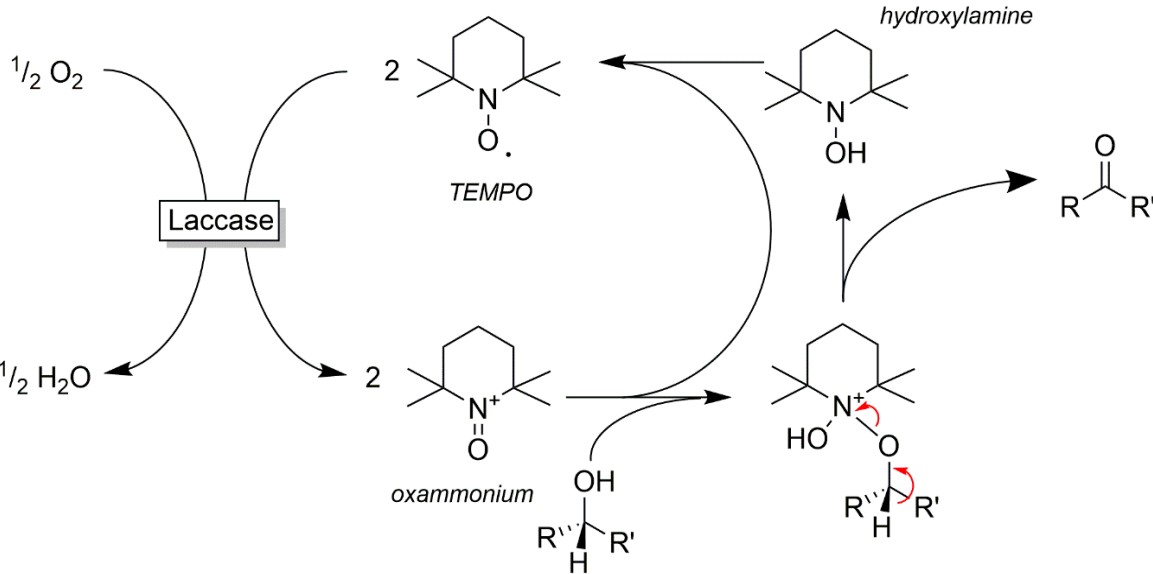

**Scheme 6.** Simplified mechanism of the LMS-mediated oxidation of alcohols.

## 2. Oxidation of Primary Alcohols

### 2.1. Oxidation of Primary Alcohols to Aldehydes

The selective oxidation of primary alcohols to the aldehyde level is generally not an issue using isolated enzymes. The catalytic mechanisms of ADHs and oxidases imply a hydride abstraction step, which precludes the ADH- or oxidase-catalyzed oxidation of aldehydes. However, if whole cell preparations are used, the presence of endogenous aldehyde dehydrogenases (*vide infra*) may impair the chemoselectivity of the oxidation reaction. In this case, the two liquid phase system (2LPS, Scheme 7) approach represents an elegant solution. A hydrophobic organic phase serves as a substrate reservoir (also enabling high reagent payloads) and simultaneously as a sink for the hydrophobic aldehyde product, extracting it from the aqueous, biocatalysts-containing phase and thereby protecting it from further oxidation to the acid.

For example, Schmid and coworkers used the two liquid phase system (2LPS) approach to control the multi-step oxidation of pseudocumene to 3, 4-dimethyl benzaldehyde [100,101]. As catalyst, recombinant *E. coli* overexpressing the xylose monooxygenase (XylM) from *Pseudomonas putida* was used [102,103]. In aqueous media, this catalyst performs through oxidation to the carboxylic acid; however, in the presence of dioctyl phthalate, the hydrophobic aldehyde intermediate preferentially partitions into the organic layer and thereby is removed from the catalyst, preventing the undesired final oxidation step.

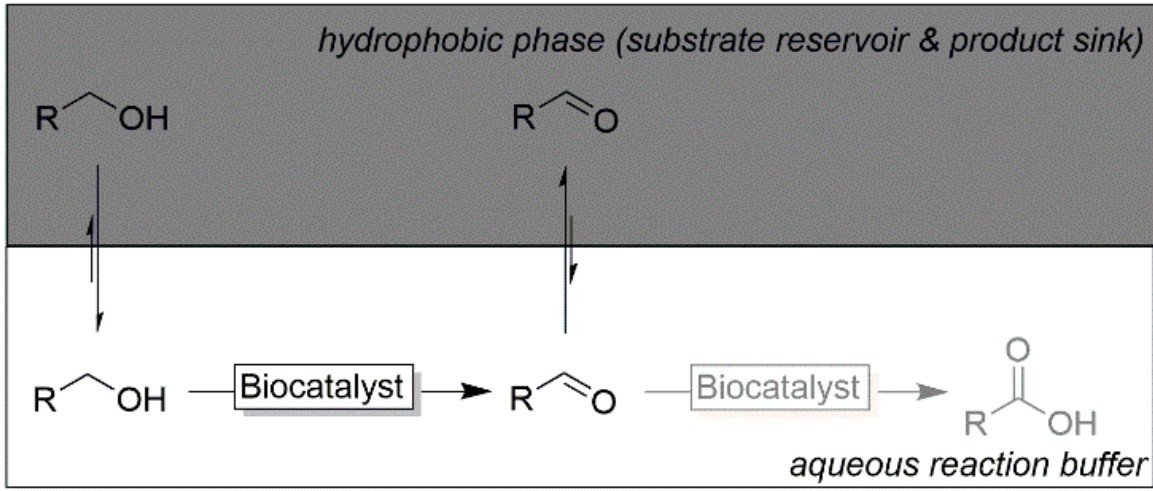

**Scheme 7.** The two liquid phase system (2LPS) approach for selective oxidation of alcohols to aldehydes.

Similarly, Molinari and coworkers have shown that the selectivity of acetic acid bacteria-catalyzed oxidation of primary alcohols can be controlled to either the aldehyde or acid level by performing the oxidation either in the presence or absence of isooctane as a hydrophobic organic phase (Table 2) [104–107].

**Table 2.** Controlling the selectivity of the whole-cell catalyzed oxidation of primary alcohols.

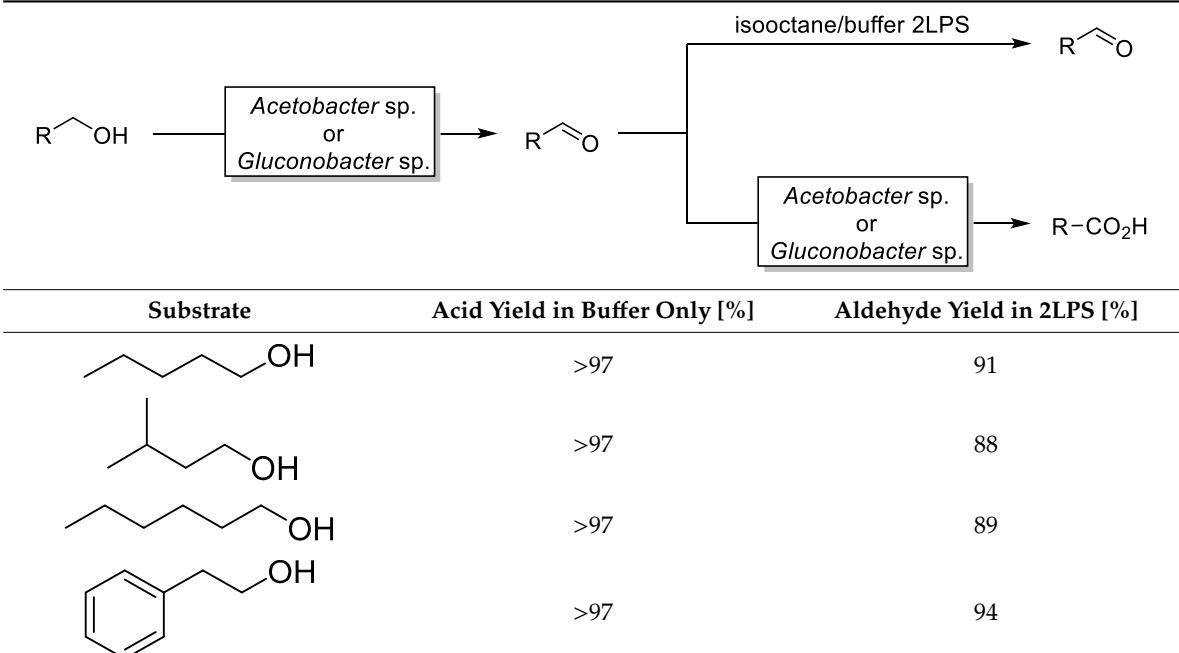

| Substrate | Acid Yield in Buffer Only [%] | Aldehyde Yield in 2LPS [%] |
|---|---|---|
| ~~~~OH | >97 | 91 |
| ~~~OH | >97 | 88 |
| ~~~~~OH | >97 | 89 |
| Ph~~OH | >97 | 94 |

The ADH-catalyzed oxidation of primary alcohols selectively to the aldehyde stage is rather scarce, even though the reaction is rather commonly used to assess ADH activity. AlcOxs are more frequently used as catalysts for the selective oxidation of primary alcohols to aldehydes.

Recently, we applied the 2LPS concept for the selective oxidation of (*2E*)-hex-2-enal into the corresponding aldehyde using the aryl alcohol oxidase from *Pleurotus eryngii* [108,109]. Product concentrations of up to 2.5 M in the organic phase and turnover numbers of the biocatalyst of more than 2 million could be achieved [110,111].

The selective oxidation of ethylene glycol to glycolaldehyde (Scheme 8) was achieved using oxidases from *Pichia pastoris* or *Aspergillus japonicus* [112,113]. Using the enzymes co-immobilized

with catalase (to eliminate $H_2O_2$), molar product concentrations were achieved. The selectivity of the reaction was very high with less than 1% of the acid overoxidation product being formed.

**Scheme 8.** Selective double oxidation of ethylene glycol to glycolaldehyde using oxidases.

In addition, the above-mentioned laccase-mediator system enjoys some popularity for the oxidation of primary alcohols [114–118]. Particularly, TEMPO is a commonly used organocatalyst enabling efficient oxidation of the alcohol starting materials. On the downside, so far, TEMPO is used in relatively high loadings (0.5–10 mol%), impairing the economic and environmental attractiveness of LMS oxidation systems.

Aldehydes are valuable reactive building blocks for further transformations. Therefore, a range of catalytic cascade reactions have been reported in which the in situ generated aldehyde is further transformed into a valuable product. Such one-pot cascades bear the advantage of circumventing at least one downstream processing, product isolation, and product purification step. This not only saves time but also resources and therefore is attractive not only from an economic but also from an environmental point of view [54].

For example, DiCosimo and coworkers reported the oxidation of glycolic acid to the corresponding aldehyde (glyoxylic acid) using methylotrophic yeasts such as *Hansenula polymorpha* or *Pichia pastoris* overexpressing spinach glycolate oxidase (GlycOx) in the presence of aminomethyl phosphonic acid (Scheme 9) [119]. The aldehyde spontaneously underwent imine formation with aminomethyl phosphonic acid, yielding N-(phosphonomethyl)glycine (glyphosate) after catalytic hydrogenation. The product was isolated and purified by simple acid precipitation and recrystallization.

**Scheme 9.** Chemoenzymatic route to produce glyphosate.

Aldehydes are also attractive reagents for further C–C bond formation reactions. For example, Siebum and coworkers combined the oxidation of pent-4-en-1-ol using a commercial AlcOx and aldolase-catalyzed aldol reaction with acetone in a one-pot two-step procedure (Scheme 10) [120]. In their proof-of-concept study, the desired β-hydroxy ketone was obtained in 30% isolated yield and 70% optical purity.

Wong and coworkers reported the combination of galactose oxidase (GalOx) with rhamnulose-1-phosphate aldolase (RhaD) to generate fructose (3 chiral centers) from simple glycerol and dihydroxy acetone phosphate (Scheme 11) [121]. An optimized reaction procedure comprising heat inactivation of the oxidase prior to aldolase addition and pH adjustment between the aldolase and the dephosphorylation step gave a respectable overall yield of 55%.

**Scheme 10.** Bienzymatic one-pot two-step cascade combining alcohol oxidase (AlcOx)-catalyzed oxidation of Pent-4-en-1-ol with an aldolase (2-desoxyribo-5-phosphate aldolase, DERA)-catalyzed aldol reaction.

**Scheme 11.** Multi-enzyme cascade combining galactose oxidase (GalOx), rhamnulose-1-phosphate aldolase, and alkaline phosphatase to synthesize fructose from glycerol and dihydroxy acetone phosphate.

More recently, Turner and coworkers used a similar cascade of an engineered galactose oxidase and rabbit muscle aldolase to produce amino sugars [122].

Acyloins become available from simple alcohols if the oxidation step is coupled to a lyase-catalyzed benzoin condensation [123–126]. The in situ generation of reactive aldehydes such as formaldehyde alleviated the toxic effect of the aldehyde on the lyase (Scheme 12) [125,126].

**Scheme 12.** Enantioselective benzoin condensation using benzaldehyde lyase (BAL) and in situ generated aldehydes.

Aldehydes can also undergo reductive amination to the corresponding amines [127,128] (Scheme 13).

**Scheme 13.** Redox-neutral cascade to transform primary alcohols into primary amines.

Finally, a recently established combination of biocatalytic (ADH- or AlcOx-catalyzed) alcohol oxidation with organocatalytic (amino acid-catalyzed) aldol condensation is worth mentioning (Scheme 14) [129–131]. Starting e.g., from 1-butanol, 2-ethyl hexenal can be obtained representing an interesting approach to upgrade bio-based alcohols.

**Scheme 14.** Upgrading of alcohols (e.g., butane-1-ol to (*2E*)-2-ethylhex-2-enal) by combining biocatalytic oxidation and organocatalytic aldol condensation.

## 2.2. Oxidation of Primary Alcohols to Acids

As mentioned above, whole cell preparations often suffer from poor selectivity if oxidation of primary alcohols to the aldehyde stage is desired. However, since through oxidation to the carboxylic acid is desired, whole cell preparations have been used from an early stage onwards. Table 3 gives a representative overview over some reported through oxidations.

**Table 3.** Selection of microbial through oxidations of primary alcohols.

| Catalyst | Product | Product Titer/Remarks | Ref. |
|---|---|---|---|
| *Acetobacter* |  | 23 gL$^{-1}$/alginate immobilized cells | [105] |
| |  R = CH$_3$ - C$_5$H$_{11}$ | 24 mM/dynamic kinetic resolution in a flow chemistry setup | [132–135] |
| |  | - | - |
| |  | Kinetic resolution | [136] |
| |  | Up to 80 g L$^{-1}$ | [137] |
| *Gluconobacter* |  | 8 gL$^{-1}$ | [138] |
| |  | Kinetic resolution | [136] |
| |  | 20 g L$^{-1}$ Kinetic resolution | [139] |
| *Corynebacterium* |  | 25 g L$^{-1}$ | [140,141] |
| *Norcardia* |  | 9 g L$^{-1}$ | [142] |

As mentioned above, ADHs and AlcOxs are generally not capable of oxidizing aldehydes, because the aldehyde proton is not abstractable as a hydride. This mechanistic limitation can be solved by nucleophilic attack to the carbonyl group transiently turning it into an alcohol containing a hydridically abstractable proton. Aldehyde dehydrogenases utilize this approach via a cysteine moiety in the enzyme active site (Scheme 15) [143].

$$R-CHO + NAD(P)^+ + H_2O \xrightarrow{\quad AldDH \quad} R-CO_2H + NAD(P)H$$

**Scheme 15.** Simplified mechanism of aldehyde dehydrogenases (AldDHs). A cysteine within the active site nucleophilically attacks the aldehyde group. The resulting hemithioacetal can transfer a hydride to the enzyme-bound oxidized nicotinamide cofactor yielding a thioester, which upon hydrolysis releases the acid product.

Preparative applications of AldDHs have been reported by several groups recently [144–147]. An early example for the through oxidation of alcohols to carboxylic acids was reported by Wong and coworkers, who combined an ADH with an AldDH for this purpose (Scheme 16) [148].

Activated aldehydes, due to a favorable aldehyde-*gem* diol equilibrium, can also be converted quite efficiently by ADHs and AlcOxs to the corresponding acids [80,149–151].

Next to water, further nucleophiles have been reported such as alcohols or amines. Especially γ- and δ-diols form hemiacetals upon aldehyde formation, which can be further oxidized to the corresponding lactones (Scheme 17) [63,68,75–77,151–157]. The hemiacetal formation is kinetically and thermodynamically favored.

**Scheme 16.** Bienzymatic cascade to transform racemic 1,2-diols or 1,2-aminoalcohols into hydroxyl- or amino acids. Due to the stereoselectivity of the ADH used, the first oxidation step proceeds as kinetic resolution.

**Scheme 17.** Oxidative lactonization of diols using the ADH from horse liver (HLADH). The intermediate aldehyde undergoes spontaneous hemiacetal formation, yielding an oxidizable hemiacetal and finally the lactone product.

Recently, also amines have caught researchers' attention as nucleophiles. For example, Turner and coworkers reported a bienzymatic cascade to transform amino alcohols into lactames via a spontaneously formed cyclic imine (Scheme 18) [122,158]. The reaction was highly pH-responsive giving higher yields at more alkaline values, which probably reflects the protonation state of the amine functionality and its tendency to nucleophilically attack the intermediate aldehyde. Similar observations have been made in case of the ADH-catalyzed oxidative lactamization [63].

**Scheme 18.** Turning amino alcohols into lactames using a cascade of Galactose oxidase (mutants) (GalOx) and periplasmic aldehyde oxidase (PaoABC).

A very interesting further development of this concept was reported recently by Mutti and coworkers [150]. By performing the GalOx-catalyzed oxidation of benzylic alcohols in the presence of ammonium buffers, they were able to obtain the corresponding nitriles in satisfactory to high yields (Scheme 19). Although a fairly broad range of alcohols could be converted in decent yields, the catalyst turnover numbers are still moderate, calling for improvement; also, the catalytic mechanism remains to be elucidated. Overall, we are convinced that this interesting reaction (and possible further cascades) will gain more attention in the near future.

The oxidation of hydroxymethyl furfural (HMF) to furan dicarboxylic acid (FDCA) has been receiving particular attention in the past years. FDCA is a potential bio-based (HMF can be obtained from glucose/fructose) substitute for terephthalic acid as building block for polyesters. Therefore, significant research efforts have been devoted to the development of biocatalytic routes to oxidize HMF to FDCA (Scheme 20). In principle, all steps of this cascade can be performed by a single oxidase

catalyst [159]. However, seemingly, the last oxidation step appears to be particularly difficult, which is why enzyme cascades are most promising (now) to attain an economically feasible full oxidation of HMF to FDCA [160,161].

**Scheme 19.** Direct conversion of primary alcohols into nitriles.

**Scheme 20.** Biocatalytic conversion of hydroxymethyl furfural (HMF) to furan dicarboxylic acid (FDCA). For reasons of simplicity, the various enzymes reported have been denominated generically as biocatalysts, and cosubstrates/coproducts have been omitted.

## 3. Oxidation of Secondary Alcohols

Compared to the reverse reaction (i.e., reduction of ketones), biocatalytic oxidations of secondary alcohols are far less common. This can be attributed to the destruction of chirality while transforming $sp^3$-hybridized alcohols into $sp^2$-hybridized carbonyl groups. Hence, value-added chiral alcohols are converted into (mostly less valuable) ketones. Nevertheless, some preparative applications of biocatalytic oxidations of secondary alcohols are known and will be discussed in the following sections.

### 3.1. Complete Oxidation of Racemic Secondary Alcohols

The complete oxidation of racemic alcohols necessitates non-stereoselective catalysts. However, non-stereoselectivity is a property seldom strived for in biocatalysis. As a consequence, identifying a suitable enzyme for the complete oxidation of racemic secondary alcohols can be a challenge.

Whole cells containing various enantiocomplementary ADHs are one option for the complete oxidation of racemic alcohols. For example, baker's yeast is principally capable of oxidizing both enantiomers of 2-heptanol [162] using two different ADHs. The expression level of both enzymes (depending on the growth phase) influenced the enantioselectivity of the *S. cerevisiae*-catalyzed oxidation and thereby makes a reproducible application difficult.

Another possibility for the complete oxidation of racemic alcohols would be to apply two enantiocomplementary biocatalysts; however, this will complicate the reaction scheme.

Ideally, non-selective ADHs would close the gap for the complete oxidation of racemic alcohols. Unfortunately, reports here are scarce (probably also because generally high enantioselectivity is desired, hence, seemingly negative results are not communicated clearly). One exception is the ADH from *Sphingobium yanoikuyae* (*Sy*ADH) reported by Kroutil and coworkers [55]. These authors purposely screened natural diversity for the non-selective oxidation of a range of racemic alcohols identifying *Sy*ADH (Scheme 21). In addition, this ADH also exhibited very high substrate tolerance, making it a very promising candidate for preparative-scale oxidations of a broad range of racemic alcohols.

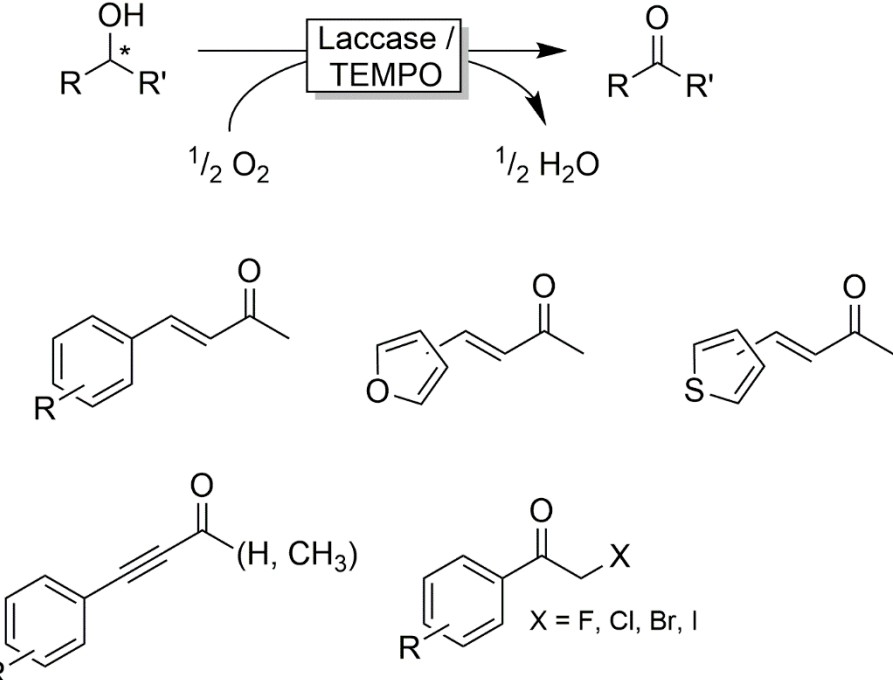

**Scheme 21.** Non-stereoselective oxidation of racemic alcohols using the ADH from Sphingobium yanoikuyae (SyADH).

In addition, an ADH from *Thermus thermophilius* may be an interesting candidate for the complete oxidation of racemic alcohols [163].

Finally, also the laccase-TEMPO system is worth mentioning here as the organocatalytic nature of the actual oxidation agent (2,2,6,6-Tetramethylpiperidin-1-yl)oxyl, TEMPO) also implies non-enantioselectivity and therefore is well-suited for the full oxidation of racemic alcohols (Scheme 22) [96,164–171].

**Scheme 22.** Laccase-TEMPO (2,2,6,6-Tetramethylpiperidin-1-yl)oxyl) system for the full oxidation of racemic alcohols.

Especially activated benzylic, allylic, or propargylic alcohols are readily converted into the corresponding ketones. Provided, the still rather low turnover numbers of the oxidation catalyst (TEMPO, ranging below 100) have been improved, this method bears some potential for the full oxidation of racemic alcohols.

### 3.2. Regioselective Oxidation of Polyols

In addition to stereoselectivity, (oxidative) enzymes also frequently exhibit regioselectivity, enabling them to perform selective transformations on poly-functionalized starting materials. Such regioselectivity is particularly interesting in the case of carbohydrate oxidations; here, a selective oxidation catalyst can avoid extensive protection and deprotection chemistry. A range of oxidases and dehydrogenases catalyzing highly regioselective oxidations of polyols are known today [172,173]. Table 4 displays some representative examples.

**Table 4.** Selection of biocatalytic oxidations of secondary alcohols.

| Product | Biocatalyst | Yield [%] | Reference |
|---|---|---|---|
| | GluOx | Up to >99 | [174] |
| | P2O | Up to >99 | [175–178] |
| | CBOx | Up to >99 | [179–183] |
| | POlDH | >99 | [184,185] |
| | AldO | >99 (10 mM) | [186–188] |

GluOx: glucose oxidase; P2O: pyranose-2-oxidase; CBOx: cellobiose oxidase; POlDH: polyol dehydrogenase; ADH-A: ADH from *Rhodococcus ruber*; ADH-9: commercial ADH; AldO: alditol oxidase from *Streptomyces coelicolor*.

Although the scope of these enzymes still is rather limited today, they exhibit a significant potential particularly in carbohydrate chemistry for protection-group independent functionalization reactions.

In the context of regioselective carbohydrate oxidation, the Reichstein process from 1934 (originally from Hoffmann–La Roche) for the transformation of glucose to ascorbic acid (vitamin C) is worth mentioning, as it still is used industrially (Scheme 23) [189].

*Gluconobacter oxydans* has also been investigated intensively for the oxidation of glycerol to dihydroxy acetone [190] to valorize the by-product from biodiesel synthesis into a building block for further chemical syntheses.

Other examples of regioselective oxidation deal with the conversion of steroids using selective hydroxysteroid dehydrogenases [191].

**Scheme 23.** Industrial pathways for the transformation of glucose to ascorbic acid.

*3.3. Kinetic Oxidative Resolution and Deracemization of Racemic Secondary Alcohols*

The stereoselective oxidation of only one alcohol enantiomer is a possibility for obtaining enantiomerically pure alcohols from racemic alcohols. To mention just one example, Kroutil and coworkers used the stereoselective ADH from *Rhodococcus ruber* (ADH-A, Scheme 24) for the kinetic resolution of a broad range of racemic alcohols [192].

**Scheme 24.** Using the ADH from *Rhodococcus ruber* (ADH-A) for the oxidative kinetic resolution of racemic alcohols.

However, kinetic resolution reactions are hampered by their intrinsic maximal yield of 50%. Deracemization reactions circumvent this drawback by recycling the unwanted ketone product back into the starting alcohol [193]. Early examples used chemical reductants such as NaBH$_3$CN yielding racemic alcohol from the ketone, which underwent further cycles of enzymatic kinetic resolution (Scheme 25a). More elegantly, Kroutil and coworkers introduced a bienzymatic reaction concept combining two enantiocomplementary ADHs wherein the first ADH catalyzes the kinetic resolution and the second ADH reduces the intermediate ketone into the desired alcohol enantiomer (Scheme 25b) [194,195].

a) deracemization via steroselecive oxidation and non-stereoselective re-reduction

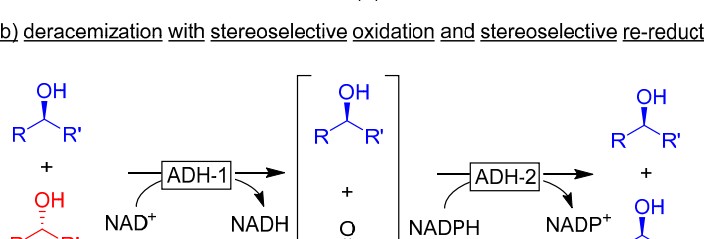

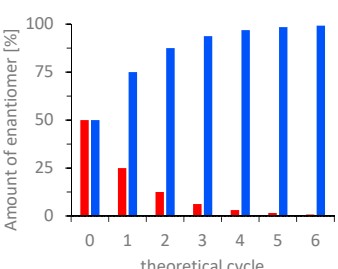

b) deracemization with stereoselective oxidation and stereoselective re-reduction

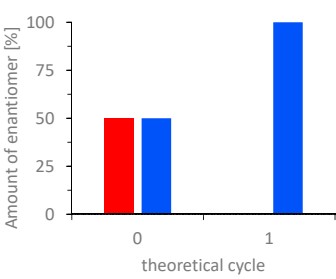

**Scheme 25.** Deracemization of alcohols combining (**a**) stereoselective kinetic resolution of the alcohol and non-selective re-reduction back into the racemate and (**b**) stereoselective oxidation combined with stereoselective re-reduction.

This principle is also applied for the stereoinversion of steroid alcohols [191,196]. The epimerization of e.g., cholic acid to chenodeoxycholic acid was possible using two enantiocomplementary hydroxysteroid dehydrogenases (Scheme 26) [196]. In contrast to the deracemization reactions described above, this reaction proceeded smoothly to almost full conversion even in the absence of any cofactor regeneration system, which was attributed to a lower energy content of the product and thereby resulting in a shifted equilibrium of the overall reaction.

**Scheme 26.** Bienzymatic epimerization of cholic acid into chenodeoxycholic acid.

## 4. Concluding Remarks

Biocatalysis offers manifold practical solutions for the oxidation of alcohols. Compared to many traditional chemical alternatives, selectivity is certainly the main feature of interest. Biocatalytic oxidation remains a very active field of research that has already solved issues such as the limited substrate scope of cofactor regeneration issues. Various promising approaches have been brought forward to solve the current issue of low substrate loadings. Hence, we are convinced that the importance of biocatalysis in alcohol oxidation will grow in the near future.

**Funding:** This research received no external funding.

**Acknowledgments:** This research was funded by the European Research Commission (ERC consolidator grant, No. 648026) and the Netherlands Organization for Scientific Research (VICI grant No. 724.014.003). Open Access Funding by TU Delft.

**Conflicts of Interest:** The authors declare no conflict of interest.

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
