# Peer review of "Biocatalytic Oxidation of Alcohols"

_catalysts, doi:10.3390/catal10090952_

Round 1

Reviewer 1 Report

This is a very relevant and timely review because the bioxidation of alcohols is undoubtely is one of the most important transformation in applied biocatalysis. This reviews is complementary to the recently published by Hollman et al. in Angwevandte Chemie (doi:10.1002/anie.202001876) and a more specialized follow-up of two older reviews about biocatalysis oxidation published in Angewandte in 2018 (doi:10.1002/anie.201800343), and in Green Chemistry in 2010 (doi:10.1039/C0GC00595A).

However, I have missed some important issues to disscuss and some inconsistencies in the description and discussion of the advances for the enzymatic oxidation of alcohol. I recommend to publish this review in this journal after addressing the following major issues.

Line 45-50, the authors talks about substrate scope, but they do not really discuss how the substrate scope can be broadened. They should discuss that genome mining, metagenomic campaigns and protein engineering have provided enzymes with a wider substrate specificity and give some paradigmatic example.

Line 52-59, the authors talks about stability and they say that “If however, the reaction procceds sufficiently fast already at more ambient temperatures, highly thermostable catalyst are not necessary”. Unfortunately, I do not agree with this argument because thermostability is linked to the longevity of the biocatalyst, and to really implement enzymes at the industrial context we need more stable enzymes although working at less drastic conditions than the natures has evolved them for. In other words, more stable enzymes will assure the high operational stability. It has proven that thermophile enzymes are highly robust during long operational runs. So, I believe that such argument should be soften.

Line 97-101. The authors discuss the mechanism of Zn2+-dependent alcohol dehydrogenases, but they forgot to mention that the first step is the deprotonation step of the hydroxyl groups, normally a base like a Tyr, a Lys or a water molecule act as base to substract the proton from the hydroxyl group to form the alcoxide intermediate, which is now ready to transfer the proton to NAD+. Many papers about the mechanism of alcohol dehydrogenases illustrate this mechanism. I just give one example: doi: 10.1016/j.jinorgbio.2017.07.022.

Line 158-169. The authors include the LMS systems for cofactor regeneration, but they do not describe this system and give any example of it below the heading. They need to explain why the laccase and its mediator is able to oxidze the NADH to NAD+ to replenish the oxidized cofactor pool required for the ADH during the alcohol oxidation. On the contrary, in this section the authors describe some of the limitation underlying the use of oxidases that come later on the next sections. For the sake of the clarity, these issues should be addressed, otherwise they mislead the reader.

Line167. The authors say that the enzymes are inactivated by liquid-gas interfaces, but unfortunately they have given some clues to solve this issues, i.e enzyme immobilization on porous carriers ( see doi:

Line 181-184. The mechanism of cooper-dependent oxidases needs to be better explained, it hard to follow it just reading the text supported with the Scheme 4. It is confusing how this enzyme uses the oxygen to oxidize the cooper, which is the one that oxidizes the alcohol through releasing H2O2.

In the section 2.1, at the end, when the authors describe the  different cascades where bioxidation of primary alcohols to aldehyde can be coupled to, they forgot to mention one exploited cascade that combines alcohol deshydrogenase and transaminase to make amines and aminoalcohols from alcohols and diols respectively. I think this cascade should be highlighted in this review. Some examples are: doi: 10.1002/anie.201204683. and 10.1002/cctc.201902404.

Line 293-297. The authors say that “ADH and AlcOX are not capable of oxidizing aldehydes because the aldehyde”….however in line 221 the authors say that “ADH-catalyzed oxidation of primary alcohols selectively to the aldehyde are rather scarce even though the reaction is rather commonly used to asses ADH activity”. From my point of view these two statements are contradictory, please could you clarify them?

Line 368-375, the authors say that non-selective ADHs are scarce, they pointed out one example from Kroutil group, here we leave another example from our group where a thermophile ADH was exploited in combination with a non-selective lipase to quantitatively transform racemic mixtures of esters into the corresponding ketone (doi: 10.1016/j.bej.2016.03.015).

In section 3.2, the selective oxidation of glycerol is missed as one of the most representative oxidation in this section. I believe that this biotransformation deserves to be mentioned because it is a well studied biooxidation that is already used at large scale; the main DHA production comes from a biocatalytic process using Gluconobacter oxidans.

Minor comments:

Line 34, the authors say that the main arguments against the use of the enzymes have been solved nowadays, I think that this argument should be softened a bit , there are still many challenges in availability, product scope, stability and cost that biocatalysis must face in the future.

Line 86-87, Pleas add a review that support this argument.

Scheme 2, The complete name of the enzyme is missed (SyADH)

Author Response

This is a very relevant and timely review because the bioxidation of alcohols is undoubtely is one of the most important transformation in applied biocatalysis. This reviews is complementary to the recently published by Hollman et al. in Angwevandte Chemie (doi:10.1002/anie.202001876) and a more specialized follow-up of two older reviews about biocatalysis oxidation published in Angewandte in 2018 (doi:10.1002/anie.201800343), and in Green Chemistry in 2010 (doi:10.1039/C0GC00595A).

Thank you very much!

However, I have missed some important issues to disscuss and some inconsistencies in the description and discussion of the advances for the enzymatic oxidation of alcohol. I recommend to publish this review in this journal after addressing the following major issues.

We have done our best to address the suggestions by reviewers 1 and 2.

Line 45-50, the authors talks about substrate scope, but they do not really discuss how the substrate scope can be broadened. They should discuss that genome mining, metagenomic campaigns and protein engineering have provided enzymes with a wider substrate specificity and give some paradigmatic example.

Changed as suggested by the reviewer.

Line 52-59, the authors talks about stability and they say that “If however, the reaction procceds sufficiently fast already at more ambient temperatures, highly thermostable catalyst are not necessary”. Unfortunately, I do not agree with this argument because thermostability is linked to the longevity of the biocatalyst, and to really implement enzymes at the industrial context we need more stable enzymes although working at less drastic conditions than the natures has evolved them for. In other words, more stable enzymes will assure the high operational stability. It has proven that thermophile enzymes are highly robust during long operational runs. So, I believe that such argument should be soften.

The reviewer should read this statement in the context of the manuscript where we compare enzymes with traditional chemical catalysts. The temperature ranges discussed here are all well beyond any enzyme operational window. Nevertheless, we have adjusted the statement.

Line 97-101. The authors discuss the mechanism of Zn2+-dependent alcohol dehydrogenases, but they forgot to mention that the first step is the deprotonation step of the hydroxyl groups, normally a base like a Tyr, a Lys or a water molecule act as base to substract the proton from the hydroxyl group to form the alcoxide intermediate, which is now ready to transfer the proton to NAD+. Many papers about the mechanism of alcohol dehydrogenases illustrate this mechanism. I just give one example: doi: 10.1016/j.jinorgbio.2017.07.022.

Changed as suggested by the reviewer.

Line 158-169. The authors include the LMS systems for cofactor regeneration, but they do not describe this system and give any example of it below the heading. They need to explain why the laccase and its mediator is able to oxidze the NADH to NAD+ to replenish the oxidized cofactor pool required for the ADH during the alcohol oxidation. On the contrary, in this section the authors describe some of the limitation underlying the use of oxidases that come later on the next sections. For the sake of the clarity, these issues should be addressed, otherwise they mislead the reader.

Changed as suggested by the reviewer.

Line167. The authors say that the enzymes are inactivated by liquid-gas interfaces, but unfortunately they have given some clues to solve this issues, i.e enzyme immobilization on porous carriers ( see doi:

Unfortunately we do not understand this comment. The references cited in the context give a good overview over the issue and some potential solutions.

Line 181-184. The mechanism of cooper-dependent oxidases needs to be better explained, it hard to follow it just reading the text supported with the Scheme 4. It is confusing how this enzyme uses the oxygen to oxidize the cooper, which is the one that oxidizes the alcohol through releasing H2O2.

Changed as suggested by the reviewer.

In the section 2.1, at the end, when the authors describe the  different cascades where bioxidation of primary alcohols to aldehyde can be coupled to, they forgot to mention one exploited cascade that combines alcohol deshydrogenase and transaminase to make amines and aminoalcohols from alcohols and diols respectively. I think this cascade should be highlighted in this review. Some examples are: doi: 10.1002/anie.201204683. and 10.1002/cctc.201902404.

Added as suggested by the reviewer.

Line 293-297. The authors say that “ADH and AlcOX are not capable of oxidizing aldehydes because the aldehyde”….however in line 221 the authors say that “ADH-catalyzed oxidation of primary alcohols selectively to the aldehyde are rather scarce even though the reaction is rather commonly used to asses ADH activity”. From my point of view these two statements are contradictory, please could you clarify them?

From our point of view both statements are valid the way they are formulated. We see no contradiction.

Line 368-375, the authors say that non-selective ADHs are scarce, they pointed out one example from Kroutil group, here we leave another example from our group where a thermophile ADH was exploited in combination with a non-selective lipase to quantitatively transform racemic mixtures of esters into the corresponding ketone (doi: 10.1016/j.bej.2016.03.015).

 Added as suggested by the reviewer.

In section 3.2, the selective oxidation of glycerol is missed as one of the most representative oxidation in this section. I believe that this biotransformation deserves to be mentioned because it is a well studied biooxidation that is already used at large scale; the main DHA production comes from a biocatalytic process using Gluconobacter oxidans.

 Added as suggested by the reviewer.

Minor comments:

Line 34, the authors say that the main arguments against the use of the enzymes have been solved nowadays, I think that this argument should be softened a bit , there are still many challenges in availability, product scope, stability and cost that biocatalysis must face in the future.

Following the reviewer’s suggestion we have adjusted this a bit. We would however refer to our section afterwards, which gives a more detailed discussion on the cost factors. We also would like to remind the reviewer that washing powder contains enzymes up to 10% w/w, hence we are tempted to state that the argument of high prices does not hold anymore today (of course there are enzymes whose production has not be scaled up yet).

Line 86-87, Pleas add a review that support this argument.

We assume the referee is referring to Figure 2. This is based on simple calculations using rule-of-thumb numbers (commonly used in the industry). To the best of our knowledge there is no review article that could be used here.

Scheme 2, The complete name of the enzyme is missed (SyADH

As far as we can see, the complete name is there.

Reviewer 2 Report

This manuscript is well organized, significant and helpful to relalated research fields.

But, small improvements are needed.

  1. The size of some Scheme and Table (ex. Scheme3, Table2, Scheme21 etc) are two small to recognize.
  2. In Table 4, the molecular structure of final product is only big.
  3. In References, Journal abbreviation should be used correctly . For example, ref. 4, 5, 10, 11, 13(also, emit some unnessary words), 20(edit should be ed.), 21, 23, 25, 27, 32, 52 and 64 (should be Catal.), 66, 68, 71, 87, 88, 96, 106, 108.....etc.

.

Author Response

  1. The size of some Scheme and Table (ex. Scheme3, Table2, Scheme21 etc) are two small to recognize.
  2. In Table 4, the molecular structure of final product is only big.
  3. In References, Journal abbreviation should be used correctly . For example, ref. 4, 5, 10, 11, 13(also, emit some unnessary words), 20(edit should be ed.), 21, 23, 25, 27, 32, 52 and 64 (should be Catal.), 66, 68, 71, 87, 88, 96, 106, 108.....etc.

Changes as suggested by the reviewer.